# On the Secondary Structure of Silk Fibroin Nanoparticles Obtained Using Ionic Liquids: An Infrared Spectroscopy Study

**DOI:** 10.3390/polym12061294

**Published:** 2020-06-05

**Authors:** Guzmán Carissimi, Cesare M. Baronio, Mercedes G. Montalbán, Gloria Víllora, Andreas Barth

**Affiliations:** 1Department of Chemical Engineering, Faculty of Chemistry, University of Murcia, 30100 Murcia, Spain; guzmanaugusto.carissimin@um.es (G.C.); mercedes.garcia@um.es (M.G.M.); gvillora@um.es (G.V.); 2Department of Biochemistry and Biophysics, Stockholm University, SE-106 91 Stockholm, Sweden; cesare.baronio@dbb.su.se

**Keywords:** silk fibroin, nanoparticles, secondary structure, infrared spectroscopy, FTIR spectroscopy, ionic liquid, curve fitting

## Abstract

Silk fibroin from *Bombyx mori* caterpillar is an outstanding biocompatible polymer for the production of biomaterials. Its impressive combination of strength, flexibility, and degradability are related to the protein’s secondary structure, which may be altered during the manufacture of the biomaterial. The present study looks at the silk fibroin secondary structure during nanoparticle production using ionic liquids and high-power ultrasound using novel infrared spectroscopic approaches. The infrared spectrum of silk fibroin fibers shows that they are composed of 58% β-sheet, 9% turns, and 33% irregular and/or turn-like structures. When fibroin was dissolved in ionic liquids, its amide I band resembled that of soluble silk and no β-sheet absorption was detected. Silk fibroin nanoparticles regenerated from the ionic liquid solution exhibited an amide I band that resembled that of the silk fibers but had a reduced β-sheet content and a corresponding higher content of turns, suggesting an incomplete turn-to-sheet transition during the regeneration process. Both the analysis of the experimental infrared spectrum and spectrum calculations suggest a particular type of β-sheet structure that was involved in this deficiency, whereas the two other types of β-sheet structure found in silk fibroin fibers were readily formed.

## 1. Introduction

Silk fibroin (SF) from the silkworm *Bombyx mori* is a fibrous protein that presents outstanding mechanical properties, which are still hard to imitate by human-made synthetic polymers [1]. The protein is produced and stored in the posterior and middle silk gland as a semi-liquid gel, in which the protein is in a water-soluble state [2] with a partially ordered structure [3], commonly known as silk I. During the spinning process, the SF secondary structure changes from semi-order to one composed mainly of antiparallel β-sheets, adopting a crystalline structure known as silk II [4,5]. This state is water-insoluble and presents all the impressive, well-known mechanical properties of silk [6]. The transition from silk I to silk II is easily achievable in the laboratory and the resulting regenerated silk can be used to produce many kinds of bioengineered materials [7,8] including but not limited to, bone fracture fixation devices [9], scaffolds for tissue engineering [10,11,12], biosensor [13,14], and nanoparticles for drug delivery applications [15,16]. The later has attracted considerable attention due to the SF nanoparticles’ (SFN) ability to load a wide variety of therapeutic compounds [17], enhance penetration [18] and the versatility of the SF matrix presents through chemical modification to program different functions into the nanoparticle [19,20,21,22]. From a biomaterial production standpoint, silk I presents an unbiased starting material for biotechnological manipulation of the properties of SF. However, obtaining native silk I directly is expensive as it is based on the direct extraction of the proteins from the worm’s gland [23] or on the purification of recombinant proteins [24]. The common alternative is using the already spun fibers (silk II) to produce silk I. Nonetheless, the production of silk I from silk II can be a challenge, as the latter presents a large network of hydrogen bonds between the SF monomer, which prevent its dissociation.

Our research group has developed a novel procedure for dissolving silk II using ionic liquids and ultrasound to produce an SF solution [25], which can be later used for the production of nanoparticles [26,27,28]. When the SF solution in ionic liquids is brought into contact with a polar organic solvent such as methanol, a desolvation process regenerates the silk II structure. This procedure represents an improvement over traditional methods [29,30] mainly because of: (i) the negligible vapor pressure and easy recyclability of ionic liquids, which make them a “greener” alternative to organic solvents [31,32,33], (ii) the possibility of obtaining high concentrations of a stable SF solution (up to 25 *w*/*w*% [26]) and (iii) the general ease with which the silk can be dissolved.

The structure of silk II is known and has been studied previously by X-ray crystallography [34], nuclear magnetic resonance (NMR) [35,36], and infrared spectroscopy [37,38]. Models for silk I structure have been presented and backed by NMR [3] and infrared [37] data. However, the structure of SF dissolved in ionic liquids and the regeneration of silk into nanoparticles from this solution have never been reported.

Many methods are available to study the structure of proteins, which all have their specific advantages and disadvantages. X-ray crystallography provides high structural resolution, but this technique requires the growth of a protein crystal that excludes its applicability to the amorphous domains of SF fibers [39]. Multidimensional NMR spectroscopy is also a powerful tool for structural determinations and its solid-state implementation makes it possible to study also large proteins like silk fibroin [36]. For its part, cryo-electron microscopy is a relatively new and powerful technique for structural determination but does not seem to have been used to obtain atomic resolutions structures of silk. Nonetheless, these methods are costly, labor-intensive, and the proteins of interest cannot be studied under all the conditions that might be of technological interest. Analysis of the circular dichroism (CD) is a useful alternative for estimating the relative amounts of secondary structures. However, β-sheet structures generate diverse CD spectra depending on the β-sheet twist, making the analysis of SF samples rather challenging [40,41]. Furthermore, samples containing large particles will scatter light and large fibers tend to precipitate, which has two consequences: (i) it leads to artifacts in the CD spectrum [42] and (ii) the less-scattering subset of the particles/fibers will dominate the CD spectrum.

Here, we chose to use attenuated total reflection Fourier transform infrared spectroscopy (ATR-FTIR), which is a versatile and cost-effective tool for determining the secondary structure of proteins [43,44,45]. Its particular advantage for the present study was that it allowed the entire nanoparticle production process to be monitored. We focused on the amide I absorption of the protein backbone (1700–1600 cm^−1^), which is mainly caused by the C=O stretching vibrations of the amide groups. The vibrations on different amide groups couple, leading to collective vibrations with vibrational frequencies that are sensitive to secondary structure [46]. Accordingly, the different secondary structures absorb in specific spectral regions [44,47,48,49,50] but the spectrum is also sensitive to further structural details like the extension and twist of β-sheets [44,51,52]. The amide I band profile is usually broad and the component bands of the different structural elements need to be identified by mathematical “resolution enhancement”—for example by calculating the second derivative, which emphasizes sharp component bands and suppresses broad ones. The such identified component bands are then used to fit the absorbance spectrum and the relative secondary structure content is evaluated from the relative band areas associated with the different secondary structures. The information on secondary structure and the accuracy of its estimation are generally similar for infrared and CD spectroscopy [53,54,55,56]—with a possible better prediction of β-sheets by infrared spectroscopy [57,58,59]. The absolute error of the secondary structure content obtained from infrared spectroscopy is mostly in the 3–10% range for a number of different evaluation methods [54,55,56,57,58,60,61,62,63,64,65].

In this work, the amide I band was analyzed by a novel method, which is based on the simultaneous fitting of the absorption spectrum and of its second derivative [66]. The method minimizes the deviation between fit and absorbance spectrum and at the same time the deviation between the second derivative of the fit and the second derivative of the experimental absorbance spectrum. This exploits more spectral information than fitting of the absorption spectrum alone because sharp component bands dominate the second derivative spectrum, whereas broad components have a stronger influence on the absorbance spectrum. Accordingly, sharp component bands are best “catched” by fitting the second derivative, whereas broad component bands are accounted for by the simultaneous fitting of the absorbance spectrum. In this way robust fit results are obtained. In addition to the experiments, calculations were carried out to simulate the infrared spectrum of different β-sheet structures within SF to confirm the band assignment. Using this approach, we studied the secondary structure of SF in fiber state (silk II) dissolved in the ionic liquid 1-ethyl-3-methylimidazolium acetate (EmimAc), and after regeneration into nanoparticle form (SFNP). The secondary structure of SF dissolved in EmimAc is hereby studied for the first time and a detailed comparison between SF fibers and regenerated SFNP is presented. Dissolved SF does not contain β-sheet structures. Upon regeneration into SFNP, two of the three types of β-sheet structures found in SF fibers are readily formed, whereas the third type is formed to a lesser extent. The structural implications are discussed.

## 2. Materials and Methods

### 2.1. Materials

EmimAc (>95% purity) was purchased from IoliTec GmbH (Heilbronn, Germany) and used without further purification. Purified water (18.2 MΩ·cm at 25 °C; from a Millipore Direct-Q1 ultrapure water system, Billerica, MA, USA) was used throughout. All other chemicals and solvents were of analytical grade and were used without further purification.

The SF used in this study was extracted from white silk cocoons of the silkworm *Bombyx mori* reared in the sericulture facilities of IMIDA (Murcia, Spain) with a diet based on fresh Morus alba L. leaves. The intact pupae were hand-extracted from the silk cocoons between two to seven days after the spinning process to avoid cross-contamination with the worm. To extract the SF, silk cocoons were shredded in a mill up to 1 mm particle size, and later boiled in 0.2 N Na_2_CO_3_ solution for 120 min to remove sericin, waxes and impurities. The remaining water-insoluble SF was rinsed thoroughly with ultrapure water and air-dried in a fume hood until constant weight.

### 2.2. Silk Fibroin Solution

The SF solution in EmimAc (SF-EmimAc) was prepared by gradually adding SF up to 10 % *w*/*w* into EmimAc under ultrasonication using a Branson 450D Sonicator (Emmerson Ultrasonic Corporation, Dansbury, CT, USA) equipped with a 12 mm diameter flat tip at 30% amplitude, in 15 s on/15 s off pulses until complete dissolution of the SF. The temperature of the solution was constantly monitored so that it did not exceed 75 °C, to avoid protein degradation. To reduce the viscosity of the SF-EmimAc solution, 3 mL of water was added to 5 g of SF-EmimAc (SF-EmimAc/H_2_O). To obtain the IR spectrum, we used D_2_O instead of H_2_O to avoid overlap between the amide I band of SF and the absorption of the H_2_O bending mode. The resulting solution is referred to as SF-EmimAc/D_2_O throughout this manuscript.

### 2.3. Preparation of Silk Fibroin Nanoparticle

SFNP were prepared following the protocol described by Lozano-Pérez et al. [26] with modifications [27]. In brief, the SF-EmimAc/H_2_O solution was heated up to 60 °C in a thermostatic bath and then sprayed over cold methanol (−20 °C) using a 0.7 mm two-fluid nozzle sprayer (from a Mini Spray DryerS B-290, BÜCHI Labortechnik, Flawil, Switzerland, Part No. 044698) and N_2_ at 1 bar as the dispersant of the aerosol. When the SF-EmimAc/H_2_O solution comes into contact with the methanol a desolvation process takes place and nanoparticles are formed. The newly formed dispersion was maintained under stirring for 2 h to allow time for a structural change. Finally, the nanoparticles were washed three times by successive centrifugation and elimination of the supernatant, and after that, freeze-dried for 72 h to obtain a fine powder.

### 2.4. Infrared Spectroscopy

Infrared absorption spectra were recorded with a Vertex 70 FTIR spectrometer (Bruker Optics, Ettlingen, Germany) continuously purged with CO_2_-free, dry air and equipped with a 1-reflection, 45° angle of incidence, diamond ATR accessory (Platinum, Bruker Optics, Ettlingen, Germany). Interferograms were recorded at a resolution of 2 cm^−1^ with a zero-filling factor of 2 and Fourier-transformed using the Blackman-Harris 3-term apodization function. Each measured spectrum was averaged from 300 scans at a data collection rate of 160 kHz. A background spectrum without sample acquired using the same number of scans before each measurement.

Spectra of statistically oriented SF fibers (<1 mm) and SFNP samples were acquired by pressing the dry protein powder with the ATR press onto the crystal. For the SF-EmimAc and SF-EmimAc/D_2_O solutions, 10 µL of the solution were placed on the ATR crystal and measured. Spectra of the ionic liquid and the ionic liquid with D_2_O were recorded and used to subtract the solvent from the SF/EmimAc and SF/EmimAc/D_2_O solutions, respectively. For all cases, the influence of the subtraction factor on the resulting amide I spectra were analyzed by using three different values (Appendix A). Spectral data were pre-processed by tracing a straight baseline from 1740 to 1560 cm^−1^ for SF fibers and SFNP and 1720 to 1600 cm^−1^ for SF/EmimAc and SF/EmimAc/D_2_O solutions. No water vapor correction was necessary.

Amino acid side chains are sometimes a concern when fitting component bands to the amide I band. However, the SF composition is skewed towards three amino acids, namely glycine (46.3%), alanine (30.8%), and serine (8.9%) [67], none of which gives a signal in the amide I region [68]. The amino acids, which do absorb in the amide I region and are present in SF are tyrosine (5.0%), glutamine (1.3%), and arginine(0.5%) [67]. The last two are in a too low concentration to have a significant effect. But Tyrosine, on the other hand, has a higher concentration and presents two absorption bands at 1590 and 1615 cm^−1^ in model compound studies [69]. Nonetheless, their molar absorptivities are rather low [63] and therefore the Tyr contribution was ignored for the analysis of the spectra.

### 2.5. Spectral Analysis and Curve-Fitting

The secondary structure of SF was analyzed by band fitting of the amide I spectrum and its second derivative simultaneously as described elsewhere [66]. Band fitting was performed using the Kinetics software written and kindly provided by Prof. Erik Goormaghtigh (ULB, Brussels) based on a MATLAB programing language and run in MATLAB version 2013b. The second derivative was smoothed with a 13-point Savitzky-Golay algorithm and multiplied by a factor of 300 to increase its weight in the fitting. The second derivative of the amide I band was used to decide the number and the initial position of the bands to be later fit by Kinetics. Bands were allowed to move in a ±10 cm^−1^ range from its initial position. The line shape of the bands used was a weighted average of a Gaussian and a Lorentzian line shape with an initial weight of 0.5 and the program was free to set any weight. The full width at half height (FWHH) of the bands was fit by the program in a range from 0 to 50 cm^−1^, starting at 25 cm^−1^. The secondary structure content was estimated from the relative band areas.

### 2.6. Calculations of Amide I Band

The calculations were performed for two different silk structures, which were based on the models provided by Asakura et al. [70] These models were extended to larger β-sheet layers using the program Mercury and manually edited in order to remove residues that stuck out from the core of the β-sheet layer. Each structure is composed of the same number of strands (32 strands) and of the same number of residues per strand (8 complete amide groups per strand). The strands are organized in 4 β-sheet layers with 8 strands each.

The simulated amide I spectra was calculated from mass-normalized force constant matrices (F matrix) using a Matlab program [71]. A mass-normalized diagonal force constant of 1.7128 mdyn Å^−1^ u^−1^, corresponding to 1705 cm^−1^, was initially assigned to each amide group. The diagonal elements of the F matrices were modified according to the effect of the local environment [72] and the effect of the inter-amide hydrogen bond, using a model suggested by Ge and coworkers [73], which is based on the Kabsch-Sander energy of hydrogen bonds [74]. The coefficients to calculate the frequency shift for the hydrogen bond to the carbonyl oxygen and the amide hydrogen were 2.4 cm^−1^/kcal and 1.0 cm^−1^/kcal, respectively. The coupling constants were obtained from density functional theory calculations for nearest neighbor interactions [72] and calculated from transition dipole coupling for other interactions. The parameters used to calculate the transition dipole coupling constants were optimized previously [75]. The magnitude of the dipole derivative was fixed at 2.20 D Å^−1^ u^−1/2^, the angle was fixed at 22° and A, a parameter that describes the effect of hydrogen bonding on the magnitude of the dipole derivative, was fixed at 0.01 cm. The position of the transition dipole moment was located 1.043 Å from the C-atom along the C=O bond and 0.513 Å along the C-N bond.

Wavenumbers and intensities of the amide I normal modes were retrieved from the solution of the eigenvalue and eigenvector problem of the F matrices [76]. The amide I spectra were calculated by assuming Gaussian lines with 16 cm^−1^ full width at half maximum for each amide I normal mode.

## 3. Results and Discussion

### 3.1. Silk Fibroin Fibers

By means of infrared spectroscopy, the *Bombyx mori* SF secondary structure was analyzed in SF fibers dissolved in EmimAc (SF-EmimAc), EmimAc and D_2_O (SF-EmimAc/D_2_O), and after regeneration into nanoparticle form (SFNP).

Results for the band fitting of the amide I band of SF fibers are presented in Figure 1, while the band parameters for all fits can be found in Table 1. The amide I band was fit with eight component bands of which the band below 1600 cm^−1^ was considered absorption from side-chain groups and was not included in the total percentage of absorbance in the amide I region. Bands at 1700 and 1692 cm^−1^ were assigned to the high wavenumber component of antiparallel β-sheets [43,77]. One band at 1678 cm^−1^ was chosen to fit the mid-high wavenumber region (1690–1660 cm^−1^) and was assigned to turn-like structures [38,61]. In the mid-region of the amide I band (1635–1660 cm^−1^), one band at 1648 cm^−1^ was used for the fit model and a perfect fit was achieved for the absorbance spectrum and a good fit for the second derivative. Even when two or three bands were used, the fit in the mid-region did not improve (data not shown). We chose to use the simplest model and assigned the area of the 1648 cm^−1^ band (33% of the entire amide I area) to irregular [78] and turn-like structures. In principle, this band could also be assigned to α-helices, the presence of which has been described previously in the mid-region of the amide I band for SF fibers in IR studies [38,79,80]. However, recent experimental data do not support the presence of α-helices in silk II [36], and thus we did not consider α-helices in our interpretation. Moreover, the large width of the band is in line with the presented assignment [78].

According to the second derivative analysis of the amide I band, the β-sheet region (ca. 1610–1640 cm^−1^, [78] Figure 1) contains at least three bands: one located at 1619 cm^−1^ corresponding to the center peak, a second one at 1609 cm^−1^ corresponding to the shoulder of the main band, and a third band that is inferred from the asymmetry of the main band at ca. 1630 cm^−1^. Note that our assignment of the 1609 cm^−1^ component band to beta sheets is different from previous assignments to Tyr [38,81] because of its low content (5%) and its relatively weak absorption [63].

Feeding the program with three bands in the low wavenumber region provided a good fit of the spectral region and its second derivative with bands centered at 1626, 1619, and 1609 cm^−1^, possibly indicating three different β-sheet structures. It should be mentioned that a poor fit was achieved for the second derivative when the 1626 cm^−1^ band was not included in the fit (data not shown). The total content of β-sheets estimated by this study was 58%, which is in line with previous IR [82], Raman-IR [83], and NMR [35] studies indicating a 50%, 50%, and 60% β-sheet content, respectively.

In the following, these three β-sheet bands are tentatively assigned to the different β-sheet structures suggested in previous studies. Silk II is composed of crystalline and non-crystalline regions, which account for 56% and 44% of the fiber, respectively. In 1999, Takahashi et al. [34] proposed a model of two different stacks of β-sheet structures present in the crystalline region. These structures are defined by the methyl group orientation of the alanines in the poly(AG) sequence within the stacked sheets, occurring statistically in a 2:1 ratio inside the crystal unit, named A and B, respectively. This model was later redefined by Asakura et al. 2002, [35] where the distances for the H-bonds were corrected. In the Takahashi model, the NH···OC interstrand hydrogen bond lengths are 2.1 Å for Ala and 2.6 Å for Gly, while in the Asakura model both distances are 1.8 Å, as determined from the same chemical shift in NMR studies [70]. Furthermore, Asakura et al. [36] proposed the existence of a third type of β-sheet, a distorted sheet in the non-crystalline fraction of silk II. Overall, their experimental evidence indicated 22% distorted β-sheets, plus 25% and 13% of the two stack models of β-sheets, giving a total β-sheet content of 60% [70]. This number is in line with our findings of 58% for the total β-sheet content. Interestingly, the band at 1626 cm^−1^ in our results accounted for 25% of the absorption in the amide I range, while the 1619 and 1609 cm^−1^ bands accounted for 9% and 18%, respectively. Because distorted β-sheets appear at higher wavenumbers than planar sheets [52], we assign the high wavenumber band to distorted sheets. The band areas of the low and mid wavenumber β-sheet bands resemble the 2:1 ratio found for structural models A and B, respectively, in the work by Asakura et al. Therefore, we tentatively assign the 1609 cm^−1^ band to structure A and the 1619 cm^−1^ band to structure B.

To our knowledge, bands have not previously been assigned to the different types of β-sheet structures of SF using ATR-FTIR spectroscopy. For this reason, we tested further the hypothesis that structure A absorbs at a lower wavenumber than structure B by spectrum calculations using the structures provided by Asakura at al. [70]. The results are shown in Figure 2, where it can be seen that structure A does indeed absorb at lower wavenumbers than structure B, supporting the tentative band assignment proposed above.

The difference between the two structures lies in the orientation of the methyl group of the alanine as represented in Figure 3, top row. On the one hand, in structure A, the methyl group points toward one Hα from Gly. On the other hand, in structure B, the methyl group points to the center of the pair of interstrand Gly···Ala hydrogen bonds. The different methyl group orientations are accompanied byslightly different backbone structures, comprising different dihedral angles, hydrogen bond lengths, and distances between the carbonyl oxygens in adjacent chains. The latter indicates a slightly different lateral position of two adjacent chains in the two structures. The distances between an oxygen in one chain and the two closest oxygens in an adjacent chain are more equal in silk B (3.6 and 4.0Å) than in silk A (3.4 and 4.1 Å) as shown in Figure 3, bottom row. This will have consequences for the amide I spectrum because the interaction with the closest amide group in the adjacent chains (connected by a short hydrogen bonded loop) leads to a downshift of the main β-sheet band, whereas that with the next closest amide group (connected by a large hydrogen bonded loop) leads to an upshift [84,85]. The distances indicate that the downshifting interaction is stronger in silk A and the upshifting interaction weaker. This expectation is confirmed by an inspection of the coupling constants determined in the course of our spectrum calculations. We conclude that the different lateral shifts of adjacent chains in silk A and B contribute to the different spectral positions of the amide I absorption maximum.

Summarizing our band assignment for SF fibers, our analysis provides excellent agreement with the NMR study by Asakura et al. [70]. We reproduce the total β-sheet content and the amount of distorted β-sheets within a few percents. We also obtain the 2:1 ratio between the occurrences of structures B and A (although the absolute content of these structures is somewhat lower than in the NMR study). This agreement validates our approach and demonstrates that it is possible to distinguish different β-sheet types in SF with infrared spectroscopy.

### 3.2. Silk Fibroin Solution in Ionic Liquid and Aqueous Ionic Liquid

The amide I band of SF dissolved in EmimAc and EmimAc/D_2_O did not show signs of β-sheet absorption as no bands were detected in the 1610–1635 cm^−1^ region (Figure 4). As was found for silk I in previous studies [86,87,88]. Fitting of the amide I band in both solvents was performed with 3 bands. In both cases, the lower wavenumber band covered two-thirds of the amide I area and the remaining third was divided between the other two bands. In the solution containing D_2_O (Figure 4b), the bands appear broader and are located at wavenumbers that are 10 cm^−1^ lower. A downshift of the same magnitude was observed when H_2_O was added (Appendix A), indicating that C=O groups are participating in a hydrogen bonding network with water [89]. The band at higher wavenumber was sharply reduced, suggesting that some structures are less stable in the presence of water or that free carbonyls become H-bonded to water and are thus red-shifted. Omitting this band caused a poor fit (data not shown).

No straightforward band assignment can be made here, as there are no previous references concerning protein secondary structures in ionic liquids. However, assignment to irregular structures and turns would be in line with conventional band assignments for proteins in an aqueous solution. Sohn et al. [90,91] suggested that the silk I state must have a higher order of structure rather than simply a combination of random structures to achieve the level of organization that silk II requires. Asakura et al. [3] proposed a repeated type-II β-turn structure for the poly AG sequence of SF in the silk I state. Further evidence of type II β-turns in the silk I structure has been obtained by infrared spectroscopy [37,92]. This structure consists of intramolecular hydrogen bonds between the carbonyl oxygen atom of the of a Gly residue at position i and an Ala residue at position i+3. Although the possible presence of an α-helix in silk I has been rejected before [36,93], this or another structure in the ionic liquid dissolution cannot be fully dismissed.

### 3.3. Silk Fibroin Nanoparticles

After dissolution of the SF in the ionic liquids and subsequent regeneration into nanoparticles, the amide I band shape of SFNP closely resembled that of silk II from SF fibers but with subtle changes that can be better depicted from its second derivative (Figure 5). In the low wavenumber range, the main band at ca. 1620 cm^−1^ was sharper and the absorbance at 1609 cm^−1^ was drastically reduced. This change can be clearly observed in the second derivative of the absorbance spectrum and could indicate that some form of β-sheet structure is reduced in the SFNP compared with that observed in SF. Moreover, the lower absorption measured at 1700 cm^−1^ is further evidence of a lower β-sheet content.

The fitting of the amide I band of SFNP yielded a similar band pattern to that seen for SF fibers (Figure 6). However, the relative area of the mid-high wavenumber band at 1674 cm^−1^ assigned to turns increased from 9% to 16%, while the relative area of the low wavenumber band at 1610 cm^−1^, tentatively assigned here to β-sheets from structure A, was reduced from 18% to 8% (Table 1). The two high wavenumber bands assigned to β-sheets were also reduced, especially the band at 1700 cm^−1^. The rest of the bands did not show any significant variation.

The formation of β-sheets from silk I can be triggered by different processes including the use of polar organic solvents [94], mechanical stress [95], and pH or ionic strength changes [96]. The process influences the structure of the final silk products as indicated by their different IR spectra in the amide I range. For example, bands assigned to β-sheets in SF films regenerated from different water annealing processes showed significantly different relative β-sheet band areas [87,97]. The intensity disparity of these component bands indicates that different regeneration protocols regenerate the β-sheet structures to different extents. The spectral variation includes a component band near 1610 cm^−1^, which has been observed previously in the IR spectra of regenerated silk [38,81,97,98]. It appears as a clear shoulder near 1610 cm^−1^ in spectra of *Antheraea assamensis* SF, when SF was regenerated with isopropanol, but is weaker when other solvents were used [81]. The observation parallels our results with *Bombyx mori* silk, where a band at a similar spectral position is weaker in regenerated SF than in natively spun SF fibers. From the similar spectral positions, we conclude that a similar type of β-sheet structure is sensitive to the regeneration process in both types of silk and we suggest that it is structure A [70].

In summary, our SFNP had 7% more turn-like structures than SF fibers and a correspondingly decreased β-sheet content. These results suggest (i) that the regeneration process of type A β-sheets was not complete, (ii) that distorted and type B β-sheet structures are preferably regenerated over type A β-sheets in the procedure used here, and (iii) that turn-like structures transition into β-sheets. This last statement is in line with previously proposed models whereby β-sheets are formed from turn-like structures [3,90].

## 4. Conclusions

Using novel infrared spectroscopic approaches, we obtained a detailed description of the secondary structure components of SF fibers (silk II), of SF dissolved in the ionic liquid, and of regenerated SFNP.

According to our analysis, SF fibers consist of 58% β-sheets, 33% irregular and/or turn-like structures, and 9% turns. The three bands at 1626, 1619, and 1609 cm^−1^ used to fit the β-sheet region were tentatively assigned to distorted sheets and the two β-sheet structures B and A [70], respectively. The band assignment was supported by simulated amide I spectra and the relative abundance of each structure is in good agreement with previous studies performed with NMR [70].

SF dissolved in ionic liquids proved to have a silk I-like structure, as characterized from its amide I band shape. This is in line with the hypothesis of the silk I structure containing irregular structures and type-II β-turns. The results demonstrate the solvation power of ionic liquids to break hydrogen bonds of hydrophobic β-structures, which has possible applications in many other fields including that of amyloid diseases [99]. Moreover, the detected loss of structure is an important observation because it presents an unbiased starting point for the biotechnological manipulation of the properties of silk, which can be used as composite material for many biomaterials.

Regenerated SFNP showed a similar amide I band shape and secondary structure composition as SF but with a 7% decrease in the β-sheet content and a corresponding increase in turns of the same magnitude. These results point to an incomplete regeneration of β-sheet structure from turn-like structures in the regeneration process using methanol. The affected β-sheet structure is suggested to be of type A, whereas the other two known β-sheet structures are formed to a similar extent as in SF fibers.

## Figures and Tables

**Figure 1 polymers-12-01294-f001:**
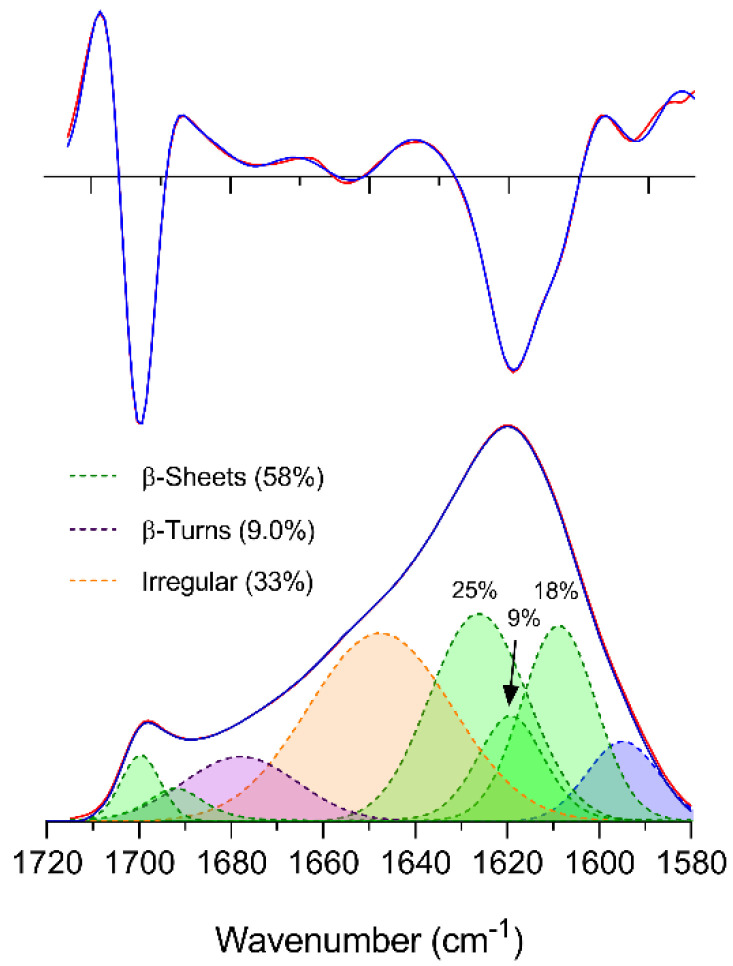
Band fitting of the amide I band and its second derivative of SF fibers. Top: second derivative of absorbance; bottom: ATR absorbance spectrum. Blue: Experimental spectrum and experimental second derivative; Red: fitted spectrum and fitted second derivative. The maximum ATR absorbance was 0.083 at 1620 cm^−1^.

**Figure 2 polymers-12-01294-f002:**
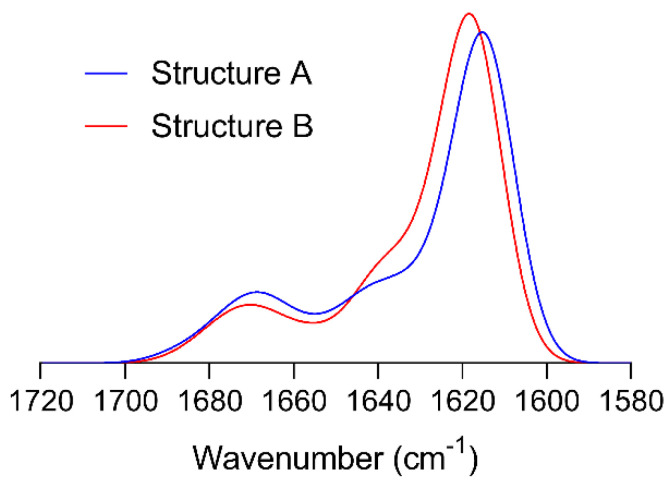
Simulated amide I spectra for the antiparallel β-sheet structures A and B [70].

**Figure 3 polymers-12-01294-f003:**
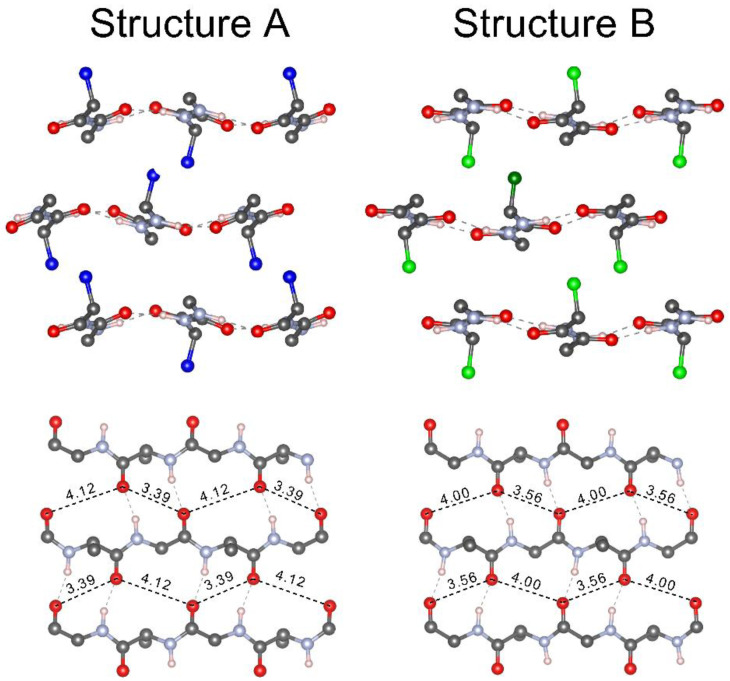
Molecular models of structures A (left column) and B (right column). Top row, different methyl orientation of structures A and B with their respective methyl groups labeled blue and green. Bottom row, distances between the closest oxygens in adjacent chains. Distances are measured in Ångström (Å) [70].

**Figure 4 polymers-12-01294-f004:**
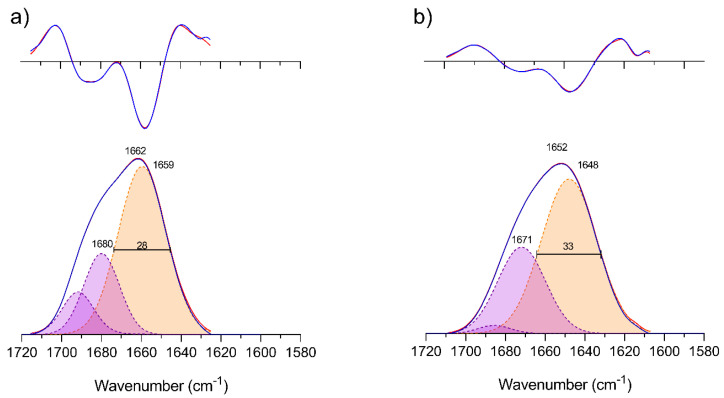
Band fitting of the amide I and its second derivative of SF dissolved in (**a**) EmimAc and (**b**) EmimAc and D_2_O after subtraction of the absorbance of the solvents. Top: second derivative of absorbance; bottom: ATR absorbance spectrum. Blue: Experimental spectrum and experimental second derivative; Red: fitted spectrum and fitted second derivative. The maximum ATR absorbance was 0.055 at 1662 cm^−1^ for panel (**a**) and 0.037 at 1652 cm^−1^ for panel (**b**).

**Figure 5 polymers-12-01294-f005:**
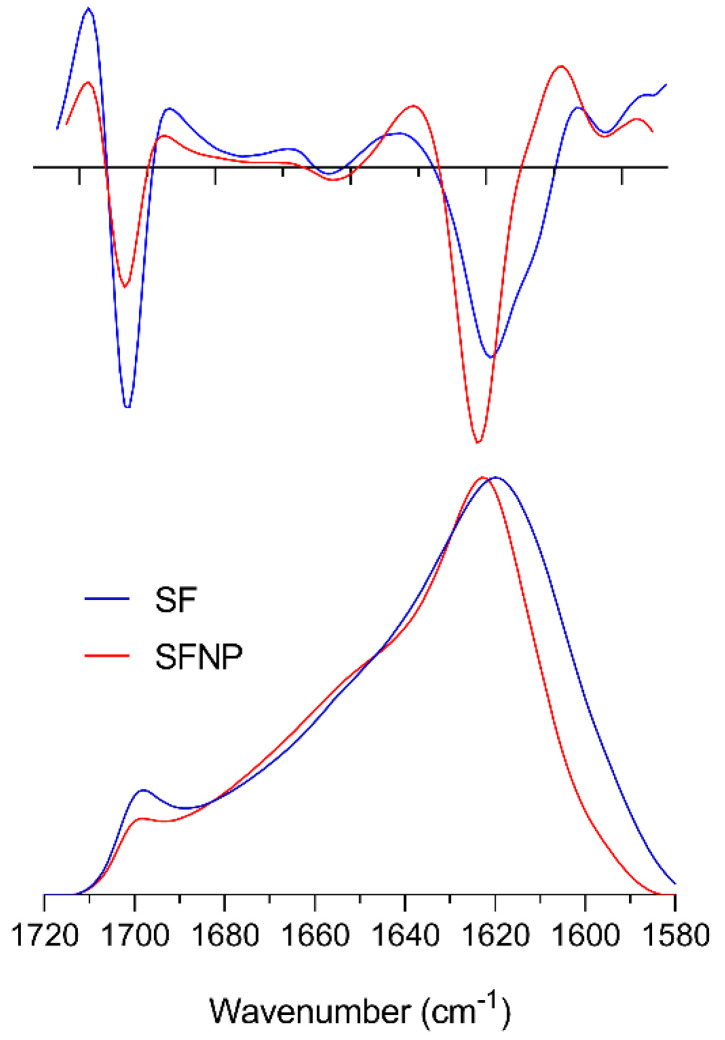
Infrared spectrum and its second derivative of SF fibers (blue) and SFNP (red). Top: second derivative of absorbance, bottom: normalized ATR absorbance spectrum of the amide I band. The maximum ATR absorbance was 0.083 at 1620 cm^−1^ for SF and 0.248 at 1622 cm^−1^ for SFNP.

**Figure 6 polymers-12-01294-f006:**
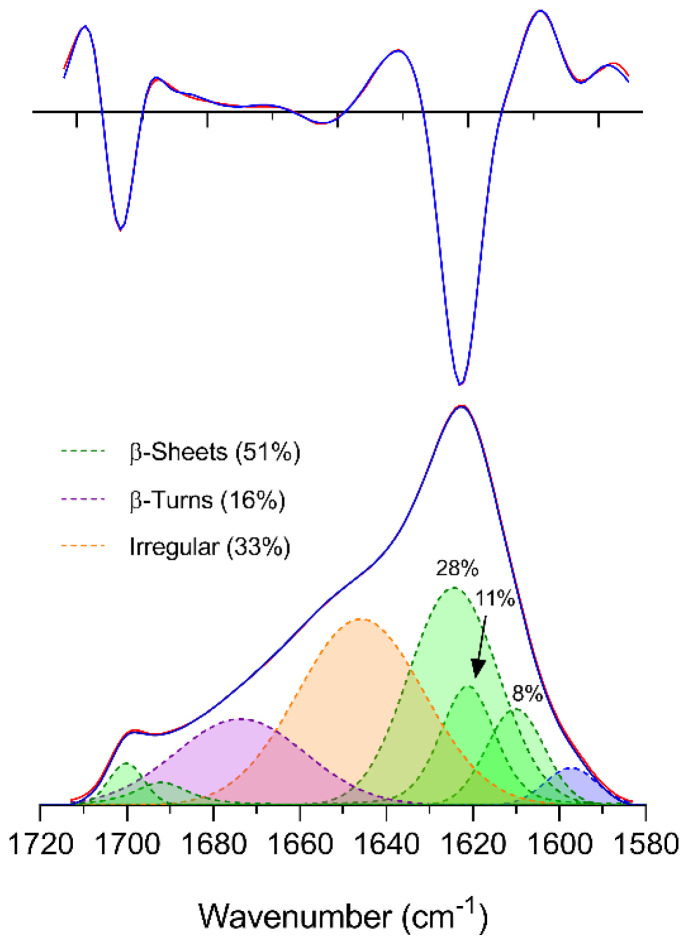
Band fitting of the amide I and its second derivative of regenerated SFNP. Top: second derivative of absorbance; bottom: ATR absorbance spectrum. Blue: Experimental spectrum and experimental second derivative; Red: fitted spectrum and fitted second derivative. The maximum ATR absorbance was 0.248 at 1622 cm^−1^.

**Table 1 polymers-12-01294-t001:** Band parameters for the fitted bands of SF fibers, SF-EmimAc, SF-EmimAc/D_2_O and SFNP.

Center (cm^−1^)	Intensity	FWHH (cm^−1^)	fg *	Relative Area (%)
**SF fibers—dry/H-Form**				
1700	0.0139	10.1	0.90	3.3
1692	0.0070	14.6	0.46	2.9
1678	0.0136	29.2	1.00	9.0
1648	0.0394	36.8	1.00	32.9
1626	0.0435	24.9	1.00	24.6
1619	0.0221	17.0	0.82	9.2
1609	0.0410	19.0	0.96	18.1
**SF-EmimAc/H-Form**				
1692	0.0134	18.8	0.85	11.7
1680	0.0253	21.1	1.00	23.3
1659	0.0526	28.4	1.00	65.0
**SF-EmimAc/D_2_O/partially deuterated**				
1686	0.0018	19.5	1.00	2.1
1672	0.0187	28.4	1.00	31.8
1648	0.0334	33.0	1.00	66.1
**SFNP—dry/H-Form**				
1700	0.0262	9.5	1.00	2.1
1692	0.0143	15.1	0.28	2.4
1674	0.0536	35.4	1.00	15.9
1646	0.1157	34.3	1.00	33.4
1624	0.1353	24.4	1.00	27.7
1621	0.0743	15.0	0.71	10.6
1610	0.0599	15.7	1.00	7.9

* The fg values denote the fractional contribution of the Gaussian lineshape to the overall lineshape. SF, silk fibroin; EmimAc, 1-ethyl-3-methylimidazolium acetate; SF-EmimAc, silk fibroin dissolved in EmimAc; SF-EmimAc/D_2_O, silk fibroin dissolved in EmimAc and heavy water; SFNP, silk fibroin nanoparticle; FWHH, full width at half height.

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
