# Peer review of "On the Secondary Structure of Silk Fibroin Nanoparticles Obtained Using Ionic Liquids: An Infrared Spectroscopy Study"

_polymers, 2020, doi:10.3390/polym12061294_

Round 1

Reviewer 1 Report

The manuscript “On the secondary structure of silk fibroin nanoparticles obtained using ionic liquids: An Infrared spectroscopy study”, is based on a study of the secondary structure of the silk fibron in its native fiber state, disolved in the ionic liquid. In the present study ATR-FTIR was used to allow the nanoparticle production process.

The study includes an interesting and complete review of the topic, where the authors of this manuscript have contributed greatly in this area.  The experimental part is also very clear. However, the results explained with a single characterization technique (ATR-FTIR) is difficult to understand the secondary structure comoponents of the fibers. It is necessary to support your results with more information or images.

Reviewer 2 Report

The article is very interesting and is very well written, being an area of ​​polysaccharide science that has advanced a lot, with the use of ionic liquids. It is only necessary to check units, for example, line 75 (degree) is misspelled. In line 90 the concentration is expressed in N, but it must be expressed in mol / L, a unit accepted by IUPAC. And if possible, it would be very interesting to add another characterization technique that complements the FTIR. However, none of this prevents the acceptance of the article.

Reviewer 3 Report

The manuscript deals with the conversion of the water-insoluble silk fibroin Silk II into the water soluble form, Silk I using ionic liquids to facilitate nanoparticle formation.  The ATFT-IR is then used to characterize the protein structure, which constitutes the core of this paper.

The paper is organized and flows smoothly; however, needs to be revised to address some of the following comments and observations:

  • The introduction needs to emphasize on the uses of Silk I and the advantages of preparation of silk nanoparticles and their application with some references.
  • In line 73, although the author meant the original state of Silk, they used to get to Silk I, nonetheless, please remove the word native to avoid confusion by the reader as traditionally, Silk I is the native state of fibroin.
  • To confirm the conclusion made in this paper and FT-IR’s ability to compare to other powerful and well-known techniques like XR-Crystallography, NMR, CD..etc in the analysis of protein structure and the changes that may occur to it, the authors should have used different concentrations of the ionic liquid to correlate that to any possible structural change and the % of that change.

Round 2

Reviewer 1 Report

The manuscript has been significantly improved and now warrants publication in Polymers.

Reviewer 3 Report

Unfortunately, the response did not address the ability of FT-IR to detect secondary structural changes in the prepared silk nanoparticles based on possible changes in preparation parameters (Concentration, temp, pH...etc)